# Integrating Sustainability in the Quality Assessment of EHEA Institutions: A Hybrid FDEMATEL-ANP-FIS Model

**Javier Puente *** , **Isabel Fernandez** , **Alberto Gomez** and **Paolo Priore**

Gijon Polytechnic School of Engineering, Department of Business Administration, University of Oviedo, 33003 Oviedo, Spain; ifq@uniovi.es (I.F.); albertogomez@uniovi.es (A.G.); priore@uniovi.es (P.P.)
* Correspondence: jpuente@uniovi.es; Tel.: +34-985181996

**Abstract:** This paper proposes the design of a conceptual model of quality assessment in European higher education institutions (HEIs) that takes into account some of the critical reflections made by certain authors in the literature regarding standards and guidelines suggested for this purpose by the European Higher Education Area (EHEA). In addition, the evaluation of the conceptual model was carried out by means of the reliable hybrid methodology MCDM-FIS (multicriteria decision making approach–fuzzy inference system) using FDEMATEL and FDANP methods (fuzzy decision-making trial and evaluation laboratory and FDEMATEL-based analytic network process). The choice of these methodologies was justified by the existing interrelationships among the criteria and dimensions of the model and the degree of subjectivity inherent in its evaluation processes. Finally, it is suggested to include sustainability as a determining factor in the university context due to its great relevance in the training of future professionals.

**Keywords:** FDEMATEL; FDANP; Fuzzy Inference System (FIS); quality assessment; EHEA; sustainability

---

## 1. Introduction

The concept of quality in European higher education is not new. Since the 1980s, national initiatives have been adopted to systematise evaluations and promote quality improvements [1,2]. *The Sorbonne Declaration* (1998), signed by the education ministers of four countries (Italy, France, Great Britain and Germany), was the prelude to the Bologna Declaration (1999), initially signed by representatives of 29 countries and subsequently extended to include 47 participants.

The Bologna Declaration laid the foundation for the construction and implementation of the so-called '*European Higher Education* Area (EHEA)', organised around four principles: quality, mobility, diversity and competitiveness. The European convergence in education aimed to promote the mobility of students, teachers and researchers while ensuring the quality of higher education, in order to ultimately increase employment in the European Union and the socioeconomic progress of the territory (http://www.eees.es/).

The objectives set by the EHEA were fundamentally articulated around the document 'Standards and guidelines for quality assurance in the EHEA' (ESG), with their monitoring and improvement discussed in subsequent meetings (Prague (2001), Berlin (2003), Bergen (2005), London (2007), Lovaina (2009), Budapest and Vienna (2010), Bucharest (2012) and Yerevan (2015)). It is important to highlight that these criteria are indicative of areas considered critical for high-quality and successful learning environments for higher education.

The criteria outlined in the ESG are classified under the following three headings: (a) internal quality assurance in higher education institutions (where 10 criteria are listed); (b) external quality

assurance in higher education (including 7 criteria); and (c) quality assurance of external quality assurance agencies (where another 7 are listed). All of the criteria are, in some way, interrelated.

Before going deeper into our study, it is appropriate to point out some reflections drawn from the literature regarding these criteria. Several authors [3,4] have pointed out three basic pillars in a university quality system: education, research and social impact. The objectives proposed for an educational system at any level (Bachelor, Master and/or PhD) will try to improve its quality and to demonstrate compliance and excellence to stakeholders [5]. This purpose requires evaluating and, if necessary, modifying those three key aspects at the unit, institutional and social level. However, as Brdulak [4] has already noted, most of the ESG criteria are widely oriented to assess and guarantee quality in the educational area, only one of the three missions assigned to the institution.

The importance of research is mentioned by the ESG standards (i.e., "The focus of the ESG is on quality assurance related to learning and teaching in higher education, including the learning environment and the relevant links for research and innovation" [6]. However, explicit guidelines are not articulated in this regard. Something similar occurs in relation to the third pillar—social impact—where the stakeholders are entities that affect or are affected by the activities of an organisation and are interested in their proper performance [7,8]. At the university level, stakeholders include a wide variety of actors from the academic community, government, students, industry, business, politics, the public sector and the general public [9]. The document mentions higher education institutions' (HEIs') commitment to community and industry stakeholders to recognise the role of these institutions "in supporting social cohesion, economic growth and global competitiveness". However, in this case, the criteria also lack the emphasis and detail reflected in the educational mission.

Apart from reflecting on the suitability of the ESG criteria to describe the three pillars of university institutions, it is necessary to consider other important issues that hinder the task of evaluating educational systems. On one hand, as indicated above, the ESG criteria are merely indicative. In fact, each country can adopt different local policies for accreditation and opt to focus on different aspects. As evidence, the criteria analysed by 15 agencies were consulted and compared. Ten of these agencies belong to the European Network for Accreditation of Engineering Education (ENAEE)—responsible for evaluating quality assurance and accreditation agencies in the EHEA in respect to their standards and procedures when accrediting engineering degrees—and five of the agencies belong to the European Registry of Quality Assurance for Higher Education (EQAR). From the comparative analysis performed, a clear consensus was detected on only 4 of the 20 factors collected to carry out the quality assessment (curriculum content; student performance and training results; material, human and financial resources; monitoring and dissemination of quality improvement measures). The other factors were generally used in a single country. On the other hand, there are different types of audits and evaluations carried out in institutions of higher education: internal audits, carried out by personnel of the assessed institutions, and external audits, carried out by teams of external experts, peers or inspectors. This variability in the formats, quality frameworks or models can generate a certain degree of scepticism about the validity of the principles of quality assessment when applied in HEIs [10]. In addition, the evaluation criteria do not focus on isolated and perfectly defined aspects; therefore, it is necessary to analyse the influence that some criteria may have on others.

Another aspect to consider is the different perception of the concept of quality by professionals, academic staff and students [11], which introduces a component of relevant subjectivity in the evaluation. This fact could be due to the elusive and multi-edge character of quality in education, as different authors have already observed, describing it as a multidimensional, multilevel and dynamic concept [12–15]. This multidimensional character includes all of the functions and activities that can be evaluated, i.e., teaching and academic programmes, research and scholarship, staff, students, facilities, equipment and services for the community and the university world, reconciling the interests of different stakeholders, national cultures and political contexts—too many dimensions to easily obtain a standardised concept of quality in HEIs.

The above reflections give an indication of the complex nature inherent in the design and implementation of a system of evaluation, control and continuous improvement of quality, not only in the teaching process but also in the whole university system, and point to the need to use a methodology capable of covering, as far as possible, the multiple facets that must be considered in the search for a solution. In this paper, two contributions are proposed: (a) the design of a conceptual model for quality assurance in European HEIs; and (b) its evaluation by means of a hybrid methodology, MCDM-FIS (multi-criterion decision-making approach–fuzzy inference system), for which no prior reference could be found in the literature.

Finally, there are many voices and agendas [16–19] articulating the need to instrumentalise higher education to train future professionals to be aware of the sustainability values. Sustainable development should be an integral and systemic part of the university institutional framework. The demand is for not only the inclusion of environmental content in the curricula of the different subjects [20], but also in the criteria for evaluation of university quality, organisation and management [21], as well as in the teacher evaluation process. The aim is to impregnate the entire university universe with the philosophy of sustainable development and the promotion of environmental education. However, it is worth noting that, despite the increasing number of universities involved with sustainability, explicit references to this issue have not been appreciated in the drafting of ESGs, a topic of great importance currently demanded for both academic and social worlds. Therefore, this work proposes as a third contribution a conceptual model to evaluate the quality of HEIs that includes an explicit dimension for sustainability.

Section 2 of this document identifies the dimensions and criteria used to define the conceptual evaluation model. In Section 3, the methodology for the development of the model through MCDM tools and FIS is detailed. Section 4 includes the empirical evaluation of the proposed model, and presents and discusses the obtained results. Finally, Section 5 presents the conclusions of the study.

## 2. Dimensions and Criteria for Quality Assurance in HEIs

In order to evaluate the proposed conceptual model, it was first necessary to identify the different dimensions to be considered in the study. This task was of critical importance to the extent that the results may be highly dependent on and sensitive to the successful selection of the most appropriate dimensions. Given the imbalance identified in linking ESGs to the three basic pillars mentioned in the previous section, it was imperative to find alternative dimensions under which to encompass the ESG.

To this aim, several possible tools to use in the continuous quality improvement field were considered (i.e., the plan–do–check–act (PDCA) tool (also referred to as the Deming cycle), Six Sigma techniques, the SERVQUAL model and the EFQM model, among others [22]). From these, the PDCA was chosen. Several reasons supported this decision. First, PDCA (plan–do–check–act) is a well-known and universally accepted tool for systematically achieving continuous quality improvement in a variety of sectors, education being one of them [23,24]. It also provides a simple but effective framework, with dimensions (shown in Figure 1) that encompass the main ESG criteria. Note that dimension 'C' brings together both check actions and their corresponding improvement activities.

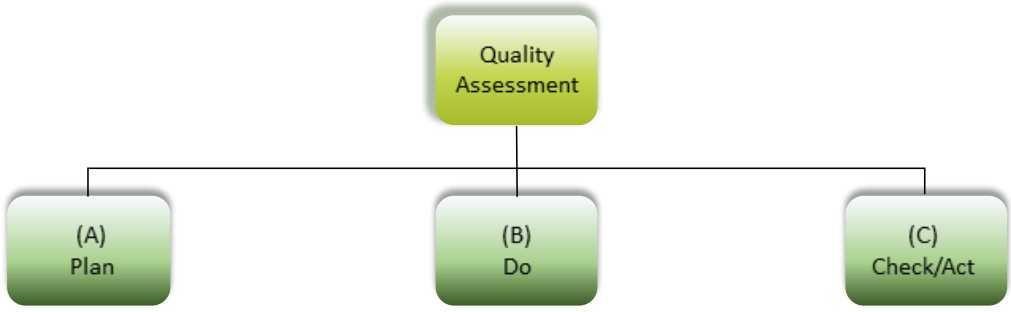

**Figure 1.** Basic dimensions of the proposed system for higher education quality assessment.

The next step involved the gathering of the ESG criteria under each of the three dimensions. As previously stated, the ESG are divided into three parts (internal quality assurance, external quality assurance and quality assurance agencies). In this paper, only the criteria included under the internal quality assurance label were considered, as self-evaluation is often carried out as a previous condition for accomplishing the external assessment [14]. In addition, the aim pursued by the authors was to demonstrate the suitability of the proposed model for quality evaluation in the higher education context; for this purpose, extending the number of criteria in order to introduce all the standards contemplated in the ESG2015 was not considered relevant.

With the focus on the first rubric, ESGs contain the following 10 criteria:

ESG01.　Policy for quality assurance
ESG02.　Design and approval of programmes
ESG03.　Student-centred learning, teaching and assessment
ESG04.　Student admission, progression, recognition and certification
ESG05.　Teaching staff
ESG06.　Learning resources and student support
ESG07.　Information management
ESG08.　Public information
ESG09.　On-going monitoring and periodic review of programs
ESG010.　Cyclical external quality assurance

As shown in Figure 2, ESG02, ESG05 and ESG06 criteria were grouped under the 'Plan' dimension, as they outline global objectives to be achieved in HEIs in terms of programmes, processes, methods and resources. Hereinafter, they are identified as A1, A2 and A3, respectively. Under the 'Do' dimension, the ESG03, ESG04 and ESG08 criteria were included, as they imply different aspects of the means of executing the previous global objectives (B1, B2, B3). Finally, the 'Check/Act' factor encompassed the ESG07, ESG09 and ESG10 criteria, which explicitly refer to the introduction of controls by which to articulate the monitoring and verification of quality assurance (C1, C2, C3). The ESG01 criterion refers to the establishment of a quality assurance policy, which captures the commitment of institutional representatives to the continuous improvement and integrity of the implemented quality systems. This criterion was not explicitly considered in the proposed model as it is a guiding principle that permeates the other PDCA dimensions.

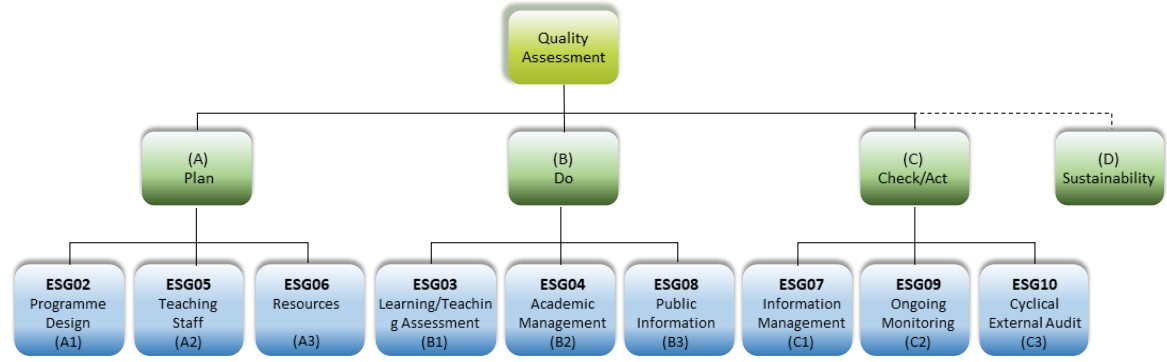

**Figure 2.** Architecture of dimensions and criteria considered in the conceptual model.

Figure 2 also depicts 'sustainability' as the fourth dimension (Dimension D) considered in the proposed conceptual model.

The methodology suggested in this paper proposes the use of the first two MCDM techniques: the Fuzzy Decision-making Trial and Evaluation Laboratory method (FDEMATEL) to identify the priorities to be assigned to the dimensions of the model, and the Analytic Network Process (ANP) to

weight the model criteria. Second, the paper proposes the design of a fuzzy inference system (FIS) that, while maintaining the structure of the conceptual model described, integrates the knowledge necessary for its evaluation.

In the next section, the MCDM and FIS methods are explained in deeper detail.

## 3. Methodology

In this paper, a systematic quality assessment system for European HEIs is proposed. The system is able to prioritise the determining factors involved in such an evaluation and to estimate the overall degree of quality of any institution based on the assessments of those factors. For this purpose, a hybrid methodology, MCDM-FIS, was chosen, for which no prior reference was found in the literature in this ambit.

The assessments were made via face-to-face interviews with eight university members (referred to, from here on, as experts). These interviews were scheduled to gather the information required for the study. The individual interviews were conducted, when possible, at interviewees' offices, and opinions on the pairwise interrelations among dimensions and criteria were requested and collected using the ad-hoc matrices prepared for that purpose. The profile of the interviewees sought required that they be members belonging to university quality committees with no less than eight years of affiliation in the institution, in order to guarantee enough knowledge and experience on the issue.

Regarding MCDM, there are many multi-criterion techniques used to obtain a criterion's weight in a decision system and to establish a ranking of its potential alternatives (Analytic Hierarchy Process –AHP-, Analytic Network Process –ANP-, Preference Ranking Organization Method for the Enrichment of Evaluations –PROMETHEE-, Decision Making Trial and Evaluation Laboratory DEMATEL, Technique for Order of Preference by Similarity to Ideal Solution –TOPSIS- and VIseKriterijumska Optimizacija I Kompromisno Resenje –VIKOR-, among others; [25–31]). Since none is clearly superior to the others [28], an appropriate combination of them should be chosen depending on the context in which they are to be applied. Fuzzy inference systems (FIS) have shown a high degree of effectiveness in the evaluation of systems when the definition of the factors on which the systems depend is subject to high degrees of uncertainty [32].

Based on the above-mentioned proposal of the conceptual evaluation model, the use of the FDEMATEL-ANP (FDANP) technique was proposed to identify the priority of the dimensions/criteria according to the influence interrelations provided by the interviewees.

Relying on the experts' opinions, the design of an FIS was then established that, while maintaining the structure of the conceptual evaluation model previously described, integrated the weights of its factors in the determination of its knowledge base.

In accordance with the above-mentioned techniques, the collaboration elicited from these university experts allowed (a) assessment of the interrelationship influences between each of the paired dimensions proposed in the model (from which the FDEMATEL method was developed); (b) assessment of the interrelationship influences among each of the paired criteria proposed in the model (from which the FDANP method will be developed); (c) proposal of the partitions of the fuzzy variables of the model inference subsystems (once the partitions were agreed, the FIS could be further designed).

The above methodologies are explained step-by-step in Section 3.1 and 3.2, and are graphically summarised in Figure 3.

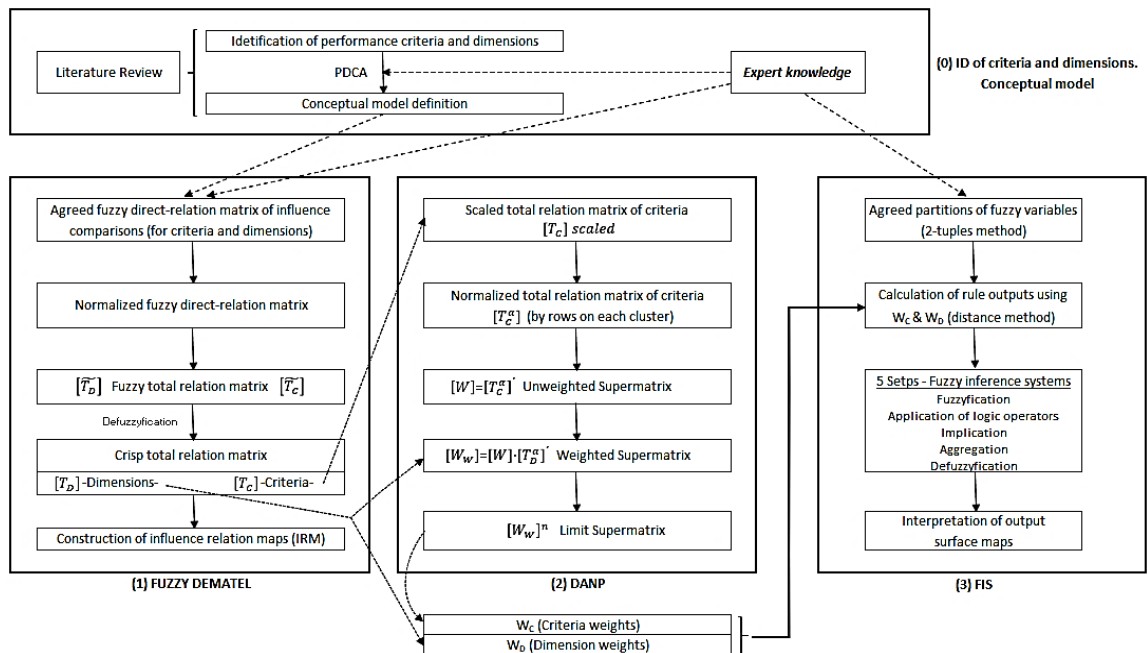

**Figure 3.** Methodology flowchart.

## 3.1. FDEMATEL-ANP (FDANP)

### 3.1.1. Fuzzy-DEMATEL

The decision-making trial and evaluation laboratory method (DEMATEL) is a multi-criterion decision technique developed in the mid-1970s by the Geneva Research Center of the Battelle Memorial Institute. DEMATEL makes it possible to convert reciprocal influence relationships between the factors (whether dimensions (D) or criteria (C)) on which a decision system depends into an intelligible structural model capable of identifying the typology of these factors (cause or effect) and assessing their impact on the system [31,33]. The representation of these impacts on maps (influence relation maps—IRM), allows the strategic factors of the decision system to be prioritised, and guides further improvement approaches. In recent years, the extension of this technique to the fuzzy field has allowed the vague assessments of experts to be processed using fuzzy numbers without modifying the general procedure of the DEMATEL technique [34]. The steps of this F-DEMATEL methodology are analytically detailed below.

Step 1. Establishment of the agreed fuzzy direct-relation matrix of pairwise influence comparisons (among '$n$' factors) $[\widetilde{A}]$ (Equation (1)).

$$[\widetilde{A}] = \begin{bmatrix} \widetilde{a}_{1,1} & \cdots & \widetilde{a}_{1,n} \\ \vdots & \ddots & \vdots \\ \widetilde{a}_{n,1} & \cdots & \widetilde{a}_{n,n} \end{bmatrix} \tag{1}$$

Every element of this matrix $\widetilde{a}_{i,j} = (l_{i,j}, m_{i,j}, u_{i,j})$ is a fuzzy number (in our case, a triangular fuzzy number—TFN) which aggregates the influence degrees—given by the group of experts—that a factor 'i' exerts on factor 'j' (i.e., through its fuzzy arithmetic mean [35]). Here, experts usually assessed the influence degrees from a fuzzy scale previously defined according to the following:

$$\left\{ \widetilde{0}_{no\ influence}, \widetilde{1}, \widetilde{2}, \widetilde{3}, \widetilde{4}_{max\ influence} \right\} = \{(0\ 0\ 1), (0\ 1\ 2), (1\ 2\ 3), (2\ 3\ 4), (3\ 4\ 4)\}$$

Step 2. Calculation of the normalised fuzzy direct-relation matrix $[\widetilde{X}]$

The matrix obtained from the previous step was normalised by parameter '*s*' (Equations (2) and (3)):

$$s = Min \left( \frac{1}{Max_{1 \leq i \leq n}(\sum_{j=1}^{n} u_{i,j})}, \frac{1}{Max_{1 \leq j \leq n}(\sum_{i=1}^{n} u_{i,j})} \right) \tag{2}$$

$$[\widetilde{X}] = s \cdot [\widetilde{A}] \tag{3}$$

Step 3. Calculation of the fuzzy total relation matrix $[\widetilde{T}]$

This matrix can be calculated by Equation (4), $[\widetilde{I}]$ being the identity matrix.

$$[\widetilde{T}] = \lim_{k \to \infty} \left( [\widetilde{X}] + [\widetilde{X}]^2 + \dots + [\widetilde{X}]^k \right) = [\widetilde{X}] \cdot [\widetilde{I} - \widetilde{X}]^{-1} \tag{4}$$

During this process, $[\widetilde{X}]$ must be decomposed into three submatrices associated with the vertexes of their fuzzy elements: $[\widetilde{X}_l]$, $[\widetilde{X}_m]$, $[\widetilde{X}_u]$ (Equation (5)).

$$[\widetilde{T}_l] = [\widetilde{X}_l] \cdot [\widetilde{I}_l - \widetilde{X}_l]^{-1} \quad [\widetilde{T}_m] = [\widetilde{X}_m] \cdot [\widetilde{I}_m - \widetilde{X}_m]^{-1} \quad [\widetilde{T}_u] = [\widetilde{X}_u] \cdot [\widetilde{I}_l - \widetilde{X}_u]^{-1} \tag{5}$$

Subsequently, upon recomposing these three matrices, $[\widetilde{T}]$ was obtained.

Step 4. Construction of the influential relation map (IRM)

Upon defuzzifying the matrix $[\widetilde{T}]$ (e.g., through the graded mean integration representation method: $a_{i,j} = \left( l_{i,j} + 4 \cdot m_{i,j} + u_{i,j} \right)/6$ [36], the total relation matrix $[T]$ was obtained. The sum of the rows and columns of this matrix allowed two vectors to be built: a column vector $[R]_{nx1}$ —sum of rows—which shows the total influence effect emanated by a factor over the rest; and a row vector $[C]_{1xn}$—sum of columns—which shows the total effect received by a factor from the rest. This row vector can be transposed to a column vector $[C]_{nx1}^T$ (Equation (6)).

$$[R] = \left[ \sum_{j=1}^{n} t_{i,j} \right]_{nx1}; \ [C] = \left[ \sum_{i=1}^{n} t_{i,j} \right]_{1xn} \to [C]^T = \left[ \sum_{i=1}^{n} t_{i,j} \right]_{nx1}^T \tag{6}$$

By adding and subtracting the homologous components of both column vectors, two parameters—associated to each factor of the system—are obtained: prominence $(R + C)$ and relation $(R - C)$. These parameters can be plotted in a two-axis chart called the IRM of the system. The prominence of a factor '*i*', $(R + C)_i$ —IRM abscissa axis—shows its strength of global influence (emanated and received); the higher the prominence, the higher the relevancy in the system. The relation of a factor '*i*', $(R - C)_i$—IRM ordinate axis—shows its net effect of influence on the system. Thus, if $(R - C)_I > 0$, it will be considered a 'cause factor' (given its influence on the rest). If instead $(R - C)_I < 0$, it will be considered an 'effect factor' (influenced by the rest). In this way, IRM helps to solve the decision-making problem by categorising all factors in the system (Figure 4).

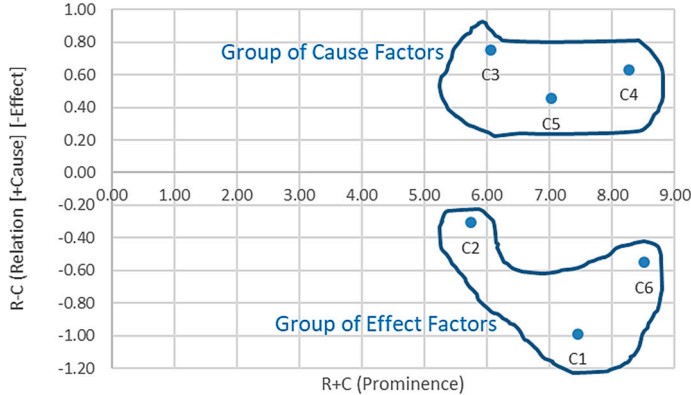

**Figure 4.** Influential relation map (IRM) example (prominence–relation).

If necessary, from the crisp total relation matrix $[T_D]$, the importance weights of these factors can be calculated by normalising that matrix according to the sum of its rows, then transposing it and, finally, raising it to high powers till convergence is reached (the latter matrix will contain by columns of identical value the weights of the factors. These dimension weights are used below in both Section 3.1.2 (to obtain the criterion weights within the ANP supermatrix) and Section 3.2 (to build the FIS rule bases).

### 3.1.2. ANP and FDANP

The analytic network process (ANP) proposed by Saaty [37] extends the analytic hierarchy process (AHP) by allowing the consideration of a network of interdependences among the different elements and levels of a decision system (objectives, criteria, sub-criteria and alternatives). Thus, although ANP is also based on pairwise comparisons, it takes into account both the dependence between the elements of a cluster (inner dependence) and between clusters themselves (outer dependence), thereby allowing more complex and realistic systems to be modelled [38]. The ANP method starts from the definition of a network of influence interdependencies (agreed a priori by the decision-makers). Subsequently, an unweighted supermatrix is built based on the priority vectors obtained from the pairwise comparison matrices between the nodes of the network (also given by the decision-makers) using some prioritisation method (usually the eigenvalue method [39]). From this matrix, a weighted supermatrix is obtained through a normalisation process (unit sum columns). Finally, by means of successive powers of this matrix, the limit supermatrix (with identical columns) is obtained, which supplies the definitive priorities of each element of the network. However, ANP usually presents problems both in defining the structural interdependencies of the network and in obtaining consistency in the judgments given by experts in matrices of pairwise comparisons. This is due to the cognitive limitations of the decision-makers and the use of nine point Likert rating scales in high-order matrices [40]. In order to avoid these problems, the total relation matrix extracted from the DEMATEL technique can be considered an unweighted supermatrix, configuring a hybrid technique called DEMATEL-based ANP (DANP), which was used in this paper. Below, the steps of this methodology in its fuzzy version (FDANP) are analytically detailed [25,41,42].

Step 1. Consideration of the total relation matrix of criteria $[T_C]$

Matrix $[T_C]$ can be obtained by defuzzifying the fuzzy total relation matrix of criteria obtained by FDEMATEL (Equation (7)), where $D^i$ denotes cluster '$i$', $C^i j$ represents criterion '$j$' within cluster '$i$' and $[Tc^{ij}]$ symbolises the relation submatrix between every two criteria from clusters '$i$' and '$j$'.

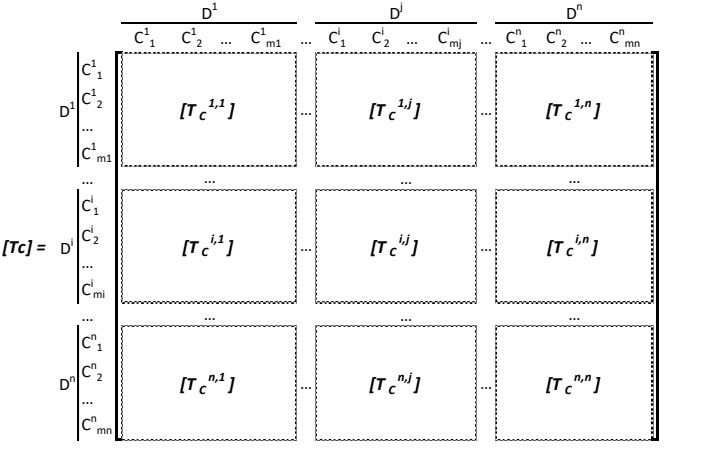

$$(7)$$

Step 2. Calculation of the normalised total relation matrix $[T_C^{\alpha}]$

$[T_C^\alpha]$ is obtained by normalising each of the submatrices $[T_C^{ij}]$ from $[T_C]$ depending on the sum of their rows. Equations (8)–(10) illustrate this process for the submatrix between Clusters 1 and 2, $[T_C^{12}]$.

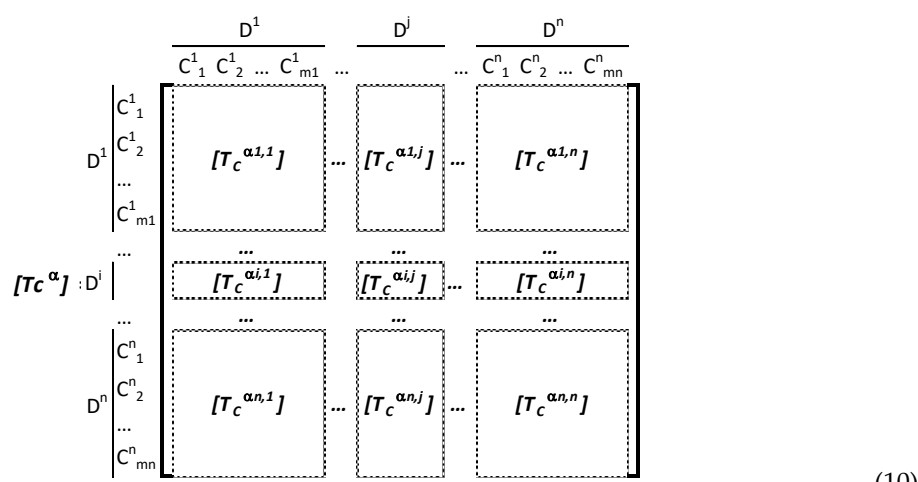

$$[Tc^{12}] = {}_{D^1}\begin{array}{c}C_1^1 \\ \cdots \\ C_i^1 \\ \cdots \\ C_{m1}^1\end{array}\begin{bmatrix} t_{1,1}^{1,2} & \cdots & t_{1,k}^{1,2} & \cdots & t_{1,m2}^{1,2} \\ \cdots & & \cdots & & \cdots \\ t_{i,1}^{1,2} & \cdots & t_{i,k}^{1,2} & \cdots & t_{i,m2}^{1,2} \\ \cdots & & \cdots & & \cdots \\ t_{m1,1}^{1,2} & \cdots & t_{m1,k}^{1,2} & \cdots & t_{m1,m2}^{1,2} \end{bmatrix}\begin{array}{l} d_1^{12} = \sum_{i=1}^{m2} t_{1,j}^{1,2} \\ d_i^{12} = \sum_{i=1}^{m2} t_{i,j}^{1,2} \\ d_{m1}^{12} = \sum_{j=1}^{m2} t_{i,j}^{1,2} \end{array}$$

(8)

$$[Tc^{\alpha 12}] = \begin{bmatrix} t_{1,1}^{1,2}/d_1^{1,2} & \cdots & t_{1,k}^{1,2}/d_1^{1,2} & \cdots & t_{1,m2}^{1,2}/d_1^{1,2} \\ \cdots & & \cdots & & \cdots \\ t_{i,1}^{1,2}/d_i^{1,2} & \cdots & t_{i,k}^{1,2}/d_i^{1,2} & \cdots & t_{i,m2}^{1,2}/d_i^{1,2} \\ \cdots & & \cdots & & \cdots \\ t_{m1,1}^{1,2}/d_{m1}^{1,2} & \cdots & t_{m1,k}^{1,2}/d_{m1}^{1,2} & \cdots & t_{m1,m2}^{1,2}/d_{m1}^{1,2} \end{bmatrix} = \begin{bmatrix} t_{1,1}^{\alpha 1,2} & \cdots & t_{1,k}^{\alpha 1} & \cdots & t_{1,m2}^{\alpha 1,2} \\ \cdots & & \cdots & & \cdots \\ t_{i,1}^{\alpha 1,2} & \cdots & t_{i,k}^{\alpha 1} & \cdots & t_{i,m2}^{\alpha 1,2} \\ \cdots & & \cdots & & \cdots \\ t_{m1,1}^{\alpha 1,2} & \cdots & t_{m1}^{\alpha 1} & \cdots & t_{m1,m2}^{\alpha 1,2} \end{bmatrix}$$

(9)

$$[Tc^\alpha] = \begin{bmatrix} [T_C^{\alpha 1,1}] & \cdots & [T_C^{\alpha 1,j}] & \cdots & [T_C^{\alpha 1,n}] \\ \cdots & & \cdots & & \cdots \\ [T_C^{\alpha i,1}] & & [T_C^{\alpha i,j}] & \cdots & [T_C^{\alpha i,n}] \\ \cdots & & \cdots & & \cdots \\ [T_C^{\alpha n,1}] & \cdots & [T_C^{\alpha n,j}] & \cdots & [T_C^{\alpha n,n}] \end{bmatrix}$$

(10)

Step 3. Obtaining the unweighted supermatrix $[W]$

$[W]$ is obtained by transposing the matrix resulting from the previous step, $[T_C^\alpha]'$ (Equation (11)).

$$[W] = [Tc^\alpha]' = \begin{bmatrix} [T_C^{\alpha 1,1}] & \cdots & [T_C^{\alpha j,1}] & \cdots & [T_C^{\alpha n,1}] \\ \cdots & & \cdots & & \cdots \\ [T_C^{\alpha 1,j}] & & [T_C^{\alpha i,j}] & \cdots & [T_C^{\alpha n,j}] \\ \cdots & & \cdots & & \cdots \\ [T_C^{\alpha 1,n}] & \cdots & [T_C^{\alpha j,n}] & \cdots & [T_C^{\alpha n,n}] \end{bmatrix}$$

(11)

Step 4. Obtaining the weighted supermatrix $[W_W]$

Once the total relation matrix of dimensions is obtained $[T_D]$, it must be normalised (by sum of rows) to give $[T_D^\alpha]$ and transposed to give $[T_D^\alpha]'$. This last matrix will act as the weighting matrix over the unweighted matrix of the previous step to obtain the weighted supermatrix $[W_W]$. This whole process is described analytically in Equations (12)–(15).

$$
[T_D] = \begin{array}{c} \\ D^1 \\ \cdots \\ D^i \\ \cdots \\ D^n \end{array}
\begin{array}{c} \begin{array}{ccccc} D^1 & \cdots & D^j & \cdots & D^n \end{array} \\
\left[ \begin{array}{ccccc}
t_{1,1}^D & \cdots & t_{1,j}^D & \cdots & t_{1,n}^D \\
\cdots & & \cdots & & \cdots \\
t_{i,1}^D & \cdots & t_{i,i}^D & \cdots & t_{i,n}^D \\
\cdots & & \cdots & & \cdots \\
t_{n,1}^D & \cdots & t_{n,j}^D & \cdots & t_{n,n}^D
\end{array} \right] \end{array}
\quad \longrightarrow \quad d_i = \sum_{j=1}^{n} t_{i,j}^D
$$

(12)

$$
[T_D^\alpha] = \begin{array}{c} \\ D^1 \\ \cdots \\ D^i \\ \cdots \\ D^n \end{array}
\left[ \begin{array}{ccccc}
t_{1,1}^D/d_1 & \cdots & t_{1,j}^D/d_1 & \cdots & t_{1,n}^D/d_1 \\
\cdots & & \cdots & & \cdots \\
t_{i,1}^D/d_i & \cdots & t_{i,j}^D/d_i & \cdots & t_{i,n}^D/d_i \\
\cdots & & \cdots & & \cdots \\
t_{n,1}^D/d_n & \cdots & t_{n,j}^D/d_n & \cdots & t_{n,n}^D/d_n
\end{array} \right]
=
\begin{array}{c} \\ D^1 \\ \cdots \\ D^i \\ \cdots \\ D^n \end{array}
\left[ \begin{array}{ccccc}
t_{1,1}^{\alpha D} & \cdots & t_{1,j}^{\alpha D} & \cdots & t_{1,n}^{\alpha D} \\
\cdots & & \cdots & & \cdots \\
t_{i,1}^{\alpha D} & \cdots & t_{i,i}^{\alpha D} & \cdots & t_{i,n}^{\alpha D} \\
\cdots & & \cdots & & \cdots \\
t_{n,1}^{\alpha D} & \cdots & t_{n,j}^{\alpha D} & \cdots & t_{n,n}^{\alpha D}
\end{array} \right]
$$

(13)

$$
[T_D^\alpha]' = \begin{array}{c} \\ D^1 \\ \cdots \\ D^j \\ \cdots \\ D^n \end{array}
\begin{array}{c} \begin{array}{ccccc} D^1 & \cdots & D^i & \cdots & D^n \end{array} \\
\left[ \begin{array}{ccccc}
t_{1,1}^{\alpha D} & \cdots & t_{i,1}^{\alpha D} & \cdots & t_{n,1}^{\alpha D} \\
\cdots & & \cdots & & \cdots \\
t_{1,j}^{\alpha D} & \cdots & t_{i,j}^{\alpha D} & \cdots & t_{n,j}^{\alpha D} \\
\cdots & & \cdots & & \cdots \\
t_{1,n}^{\alpha D} & \cdots & t_{i,n}^{\alpha D} & \cdots & t_{n,n}^{\alpha D}
\end{array} \right] \end{array}
$$

(14)

$$
[W_W] = \begin{array}{c} \\ D^1 \\ \\ \\ D^j \\ \\ D^n \\ \\ \end{array}
\begin{array}{c}
\begin{array}{ccc} \quad\quad D^1 \quad\quad & \quad D^i \quad & \quad\quad D^n \quad\quad \end{array} \\
\begin{array}{ccc} C_1^1\ C_2^1\ \cdots\ C_{m1}^1 & \cdots & C_1^n\ C_2^n\ \cdots\ C_{mn}^n \end{array} \\
\left[ \begin{array}{ccc}
[W^{11}]\cdot t_{1,1}^{\alpha D} & \cdots\ [W^{i1}]\cdot t_{i,1}^{\alpha D}\ \cdots & [W^{n1}]\cdot t_{n,1}^{\alpha D} \\
\cdots & \cdots & \cdots \\
[W^{1j}]\cdot t_{1,j}^{\alpha D} & [W^{ij}]\cdot t_{i,j}^{\alpha D}\ \cdots & [W^{nj}]\cdot t_{n,j}^{\alpha D} \\
\cdots & \cdots & \cdots \\
[W^{1n}]\cdot t_{1,n}^{\alpha D} & \cdots\ [W^{in}]\cdot t_{i,n}^{\alpha D}\ \cdots & [W^{nn}]\cdot t_{n,n}^{\alpha D}
\end{array} \right]
\end{array}
$$

(15)

Finally, by raising $[W_W]$ to high powers, a limit supermatrix of identical columns is obtained. The values in those columns correspond to the definitive weights of all criteria. These criterion weights are used (in conjunction with dimension criteria obtained in Section 3.1.1) to build the FIS rule bases, as explained in Section 3.2.

### 3.2. Fuzzy Inference Systems

Fuzzy inference systems (FIS) allow the incorporation of expert decision-support knowledge in the form of fuzzy rules that relate system variables subject to subjective or uncertain assessments (which are difficult to accurately describe using traditional numerical variables). To design an FIS, its

knowledge base must initially be defined as follows: (a) linguistic labels that determine the partitions of its fuzzy input and output variables; and (b) 'if–then' type rules reflecting the label to be given to each output fuzzy variable according to the label set of the input fuzzy variables (e.g., IF [*Var1* is LOW & *Var2* is MED] THEN [*Var3* is MED]). In the processing of these fuzzy variables, the use of trapezoidal fuzzy numbers has been demonstrated to have sufficient efficiency and utility [43].

Subsequently, the process of inference (Mamdani type) takes place in five phases: (1) crisp input fuzzification of each of the cases to infer; (2) application of fuzzy operators in the antecedent of each rule; (3) implication to the consequent of each rule; (4) aggregation of all the implicated consequents of the entire rule base; and (5) defuzzification of the final aggregate. A detailed description of each of these phases can be found in [44]).

In general terms, the definition of the knowledge base is built upon the judgement of a team of experts who agree on: (a) the typology of the labels to be used in all variables as well as their corresponding partitions; and (b) the labels to be assigned to the output of each rule, which will subsequently trigger the inference process.

In this paper, to partition the variables, the expert team relied on a linguistic representation method based on 2-tuples that allowed its systematic design by means of fuzzy numbers. To this aim, and based on a linguistic preferences scale previously defined, the parameterisation of all partitions for all system variables was agreed [45,46].

Regarding the definition of the system rule based on the model, it was decided to identify the label to be assigned to each rule by the minimum distance from the weighted fuzzy sum of its input labels (using as local weights those obtained using the FDANP methodology) to all labels in the agreed partition of the output variable. The use of a knowledge elicitation method based on the extension principle [47] according to the expression given in Equation (16) was proposed to be used in the calculation of these distances, assuming trapezoidal fuzzy numbers:

$$
\begin{aligned}
Dj &= D\big([a_o, b_o, c_o, d_o] - [a^j, b^j, c^j, d^j]\big) \\
&= \sqrt{P_a\big(a_o - a^j\big)^2 + P_b\big(b_o - b^j\big)^2 + P_c\big(c_o - c^j\big)^2 + P_d\big(d_o - d^j\big)^2}
\end{aligned}
\tag{16}
$$

where $[a_o, b_o, c_o, d_o]$ represents the weighted fuzzy output of the rule for which the output is to be estimated, $[a^j, b^j, c^j, d^j]$ is the *j-th* fuzzy label of the output variable partition, and *Pa*, *Pb*, *Pc* and *Pd* are the priorities given to the vertices of fuzzy trapezoids considered (i.e., $Pa + Pb + Pc + Pd = 1$).

Finally, the label of the partition to which the previous distance was minimal was chosen as the output label. This methodological proposal is feasible if the input and output variables have identical ranges and label progression (lower values–worse/higher values–better). In this way, the procedure exempts the expert team from agreeing to the outputs of the rules, which results both in the prevention of judgement inconsistency with respect to influence valuations issued on FDANP and in a substantial resource saving in the rule extraction process.

In this paper, the design of the model knowledge base was implemented using the Matlab Fuzzy Logic Toolbox ©. The inference maps available in this tool allow the analysis, in a simple and intuitive manner, of the evaluations obtained in any HEI. Thus, each map evaluated an output variable for any values of two of its input variables—those shown on the map—and fixed values of the rest.

## 4. Empirical Evaluation of the Proposed Quality Assessment Model: Results and Discussion

In this section, the above-explained methodologies were applied to the proposed quality assessment model.

### 4.1. F-DEMATEL

Initially, the eight expert team members provided pairwise evaluations on the interrelationship influences among the four dimensions of the proposed model (A—'plan', B—'do', C—'check/act' and D—'sustainability'). They used a five level linguistic preference scale (triangular fuzzy numbers—TFN).

$$\left\{ \widetilde{0}_{no\ influence},\ \widetilde{1},\ \widetilde{2},\ \widetilde{3},\ \widetilde{4}_{max\ influence} \right\} = \{(0\ 0\ 1),\ (0\ 1\ 2),\ (1\ 2\ 3),\ (2\ 3\ 4),\ (3\ 4\ 4)\}$$

The agreed fuzzy direct-relation matrix of pairwise influence comparisons $[\widetilde{A}]$ (Equation (17)) was obtained by using the fuzzy arithmetic mean [35] of those evaluations; from it, the steps outlined in Section 3.1.1 were followed.

| $[\widetilde{A}]$ | A | | | B | | | C | | | D | | |
|---|---|---|---|---|---|---|---|---|---|---|---|---|
| **A** | 0.00 | 0.00 | 0.00 | 2.38 | 3.38 | 3.88 | 1.13 | 2.13 | 3.13 | 0.88 | 1.88 | 2.88 |
| **B** | 0.13 | 0.75 | 1.75 | 0.00 | 0.00 | 0.00 | 2.38 | 3.38 | 3.88 | 1.38 | 2.38 | 3.38 |
| **C** | 0.63 | 1.63 | 2.63 | 0.25 | 1.25 | 2.25 | 0.00 | 0.00 | 0.00 | 0.25 | 1.25 | 2.25 |
| **D** | 0.50 | 1.50 | 2.50 | 0.00 | 1.00 | 2.00 | 0.00 | 0.50 | 1.50 | 0.00 | 0.00 | 0.00 |

(17)

After normalising the $[\widetilde{A}]$ matrix with s = 9,88 (Equations (2)–(5)), the fuzzy total relation matrix of the four dimensions $[\widetilde{T_D}]$ (Equation (18)) was estimated.

| $[\widetilde{T_D}]$ | A | | | B | | | C | | | D | | |
|---|---|---|---|---|---|---|---|---|---|---|---|---|
| **A** | 0.02 | 0.17 | 0.92 | 0.25 | 0.50 | 1.32 | 0.18 | 0.44 | 1.32 | 0.13 | 0.40 | 1.31 |
| **B** | 0.04 | 0.22 | 0.98 | 0.01 | 0.18 | 0.92 | 0.25 | 0.47 | 1.25 | 0.15 | 0.38 | 1.23 |
| **C** | 0.07 | 0.25 | 0.92 | 0.04 | 0.26 | 0.99 | 0.02 | 0.15 | 0.83 | 0.04 | 0.25 | 1.02 |
| **D** | 0.05 | 0.21 | 0.82 | 0.01 | 0.21 | 0.87 | 0.01 | 0.17 | 0.87 | 0.01 | 0.11 | 0.74 |

(18)

This matrix was then defuzzified by using the graded mean integration representation method to compute the crisp total relation matrix $[T_D]$ (Equation (19)). From this, once the dimensions' prominences (**R + C**) and relations (**R − C**) (according to Equation (6)) were calculated, the influential relation map (Figure 5) of the model dimensions was plotted.

| $[T_D]$ | A | B | C | D | R | C | R + C | R − C |
|---|---|---|---|---|---|---|---|---|
| **A** | 0.27 | 0.59 | 0.54 | 0.51 | 1.91 | 1.21 | 3.12 | 0.70 |
| **B** | 0.32 | 0.27 | 0.56 | 0.49 | 1.64 | 1.49 | 3.14 | 0.15 |
| **C** | 0.33 | 0.34 | 0.24 | 0.35 | 1.26 | 1.61 | 2.88 | -0.35 |
| **D** | 0.29 | 0.29 | 0.26 | 0.20 | 1.03 | 1.54 | 2.57 | -0.50 |

(19)

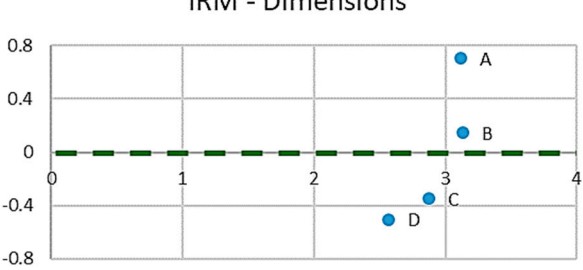

**Figure 5.** Influential relation map (four dimensions).

The chart shows that the dimensions A—'plan' and B—'do' had the most important prominences and were cause-type dimensions, whereas dimensions C—'check/act' and D—'sustainability' were effect-type dimensions. This result strongly suggests, in line with the real world, that the sustainability dimension is influenced by the planning and implementation dimensions of the model, and that measures and procedures taken in the two early phases of the quality assessment system will translate into clear impacts upon it.

From the matrix $[T_D]$, the importance weights of each dimension $[W_{D4}]$ (see the shadow column in Equation 20) were calculated by first normalising the matrix according to the sum of its rows to give $[T_{D4}^{\alpha}]$, then transposing it to give $[T_{D4}^{\alpha}]'$ and, finally, raising it to high powers until convergence was reached at $\left([T_{D4}^{\alpha}]'\right)^{n}$.

| $[T_{D4}^{\alpha}]$ | A | B | C | D | $[T_{D4}^{\alpha}]'$ | A | B | C | D | $\left([T_{D4}^{\alpha}]'\right)^{n}$ | A | B | C | D |
|---|---|---|---|---|---|---|---|---|---|---|---|---|---|---|
| A | 0.14 | 0.31 | 0.28 | 0.26 | A | 0.14 | 0.19 | 0.26 | 0.28 | A | **0.22** | 0.22 | 0.22 | 0.22 |
| B | 0.19 | 0.17 | 0.34 | 0.30 | B | 0.31 | 0.17 | 0.27 | 0.28 | B | **0.26** | 0.25 | 0.25 | 0.25 |
| C | 0.26 | 0.27 | 0.19 | 0.27 | C | 0.28 | 0.34 | 0.19 | 0.25 | C | **0.27** | 0.27 | 0.27 | 0.27 |
| D | 0.28 | 0.28 | 0.25 | 0.19 | D | 0.26 | 0.30 | 0.27 | 0.19 | D | **0.26** | 0.26 | 0.26 | 0.26 |
|  |  |  |  |  |  |  |  |  |  |  | **W$_{D4}$** |  |  |  |

$$(20)$$

The results showed that sustainability (Dimension D) obtained a substantial weighting in the system (26%). This result endorsed some statements from the literature about the potential responsibility of HEIs in bringing about necessary changes not only through research, but also through better training of their graduates, those who are becoming professionals capable of responding more accurately to the challenges posed by the sustainability paradigm [48,49]. This significant figure of 26% suggests, among other things, the need to make efforts to reach a consensus, for instance, on the standardisation of certain sustainability criteria in order to incorporate them into assessment models.

Based on the same evaluations provided by the expert team, but exclusively considering the traditional three dimensions (**A**, **B** and **C**), the weights $[W_{D3}]$ would have been those given in Equation (21). In this alternative scenario, the preponderance of dimension C—'check/act' stood out over A—'plan' and B—'do'. These were the data used in the FDANP methodology to determine the overall weighting of the criteria on which these three dimensions depend.

| $[T_{D3}^{\alpha}]$ | A | B | C | $[T_{D3}^{\alpha}]'$ | A | B | C | $\left([T_{D3}^{\alpha}]'\right)^{n}$ | A | B | C |
|---|---|---|---|---|---|---|---|---|---|---|---|
| A | 0.18 | 0.41 | 0.41 | A | 0.18 | 0.25 | 0.32 | A | **0.26** | 0.26 | 0.26 |
| B | 0.25 | 0.24 | 0.51 | B | 0.41 | 0.24 | 0.37 | B | **0.34** | 0.34 | 0.34 |
| C | 0.32 | 0.37 | 0.31 | C | 0.41 | 0.51 | 0.31 | C | **0.40** | 0.40 | 0.40 |
|  |  |  |  |  |  |  |  |  | **W$_{D3}$** |  |  |

$$(21)$$

### 4.2. FDANP

After averaging the pairwise fuzzy allocations given by experts to the influence interrelationships among the model criteria, a $9 \times 9$ fuzzy matrix was obtained. After defuzzifying it and scaling it in the range of scores 0–4 [33], the total relation matrix of criteria $[T_C]$ (Equation (22)) was obtained. From

this matrix, and once the corresponding prominences (**R + C**) and relations (**R − C**) were calculated, the influential relation maps (Figure 6) of the model criteria were determined.

| $[T_C]$ | A1 | A2 | A3 | B1 | B2 | B3 | C1 | C2 | C3 | Row Sums | | | R | C | R+C | R-C |
|---|---|---|---|---|---|---|---|---|---|---|---|---|---|---|---|---|
| A1 | 1.20 | 1.50 | 2.30 | 3.72 | 1.31 | 2.28 | 2.65 | 2.75 | 2.47 | 4.99 | 7.30 | 7.87 | 20.17 | 17.13 | 37.29 | 3.04 |
| A2 | 2.01 | 0.84 | 1.66 | 3.51 | 1.65 | 1.61 | 2.46 | 2.56 | 1.83 | 4.51 | 6.78 | 6.84 | 18.13 | 14.11 | 32.24 | 4.02 |
| A3 | 1.51 | 1.82 | 1.12 | 3.50 | 1.68 | 2.06 | 2.45 | 2.09 | 1.80 | 4.45 | 7.24 | 6.34 | 18.04 | 17.70 | 35.73 | 0.34 |
| B1 | 1.49 | 1.56 | 2.35 | 1.88 | 1.33 | 2.29 | 3.46 | 3.26 | 2.95 | 5.41 | 5.50 | 9.67 | 20.57 | 25.81 | 46.38 | -5.23 |
| B2 | 0.30 | 0.59 | 0.74 | 1.04 | 0.00 | 0.73 | 0.90 | 0.97 | 0.81 | 1.63 | 1.77 | 2.69 | 6.09 | 12.08 | 18.17 | -5.99 |
| B3 | 2.25 | 1.99 | 1.83 | 2.88 | 1.81 | 1.29 | 2.60 | 3.20 | 2.44 | 6.07 | 5.98 | 8.24 | 20.28 | 17.46 | 37.74 | 2.83 |
| C1 | 3.48 | 2.31 | 3.16 | 3.48 | 1.62 | 2.23 | 2.07 | 4.00 | 3.29 | 8.95 | 7.33 | 9.37 | 25.65 | 21.78 | 47.43 | 3.87 |
| C2 | 2.73 | 2.02 | 2.33 | 2.99 | 1.38 | 2.31 | 2.65 | 1.80 | 2.48 | 7.08 | 6.68 | 6.93 | 20.69 | 23.29 | 43.98 | -2.60 |
| C3 | 2.16 | 1.47 | 2.21 | 2.80 | 1.29 | 2.67 | 2.52 | 2.66 | 1.39 | 5.85 | 6.76 | 6.58 | 19.19 | 19.46 | 38.65 | -0.27 |

(22)

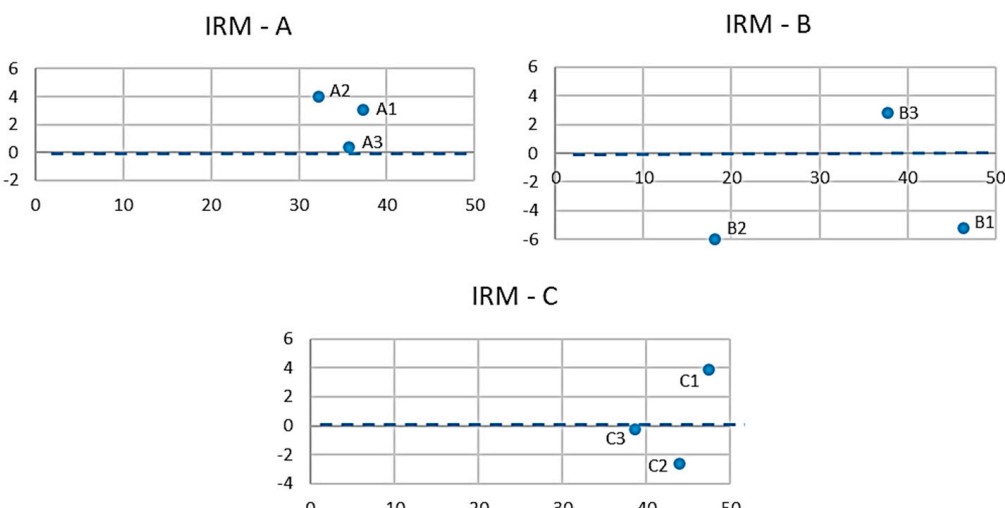

**Figure 6.** Influential relation maps (nine criteria).

Figure 6 shows that all criteria had considerable prominence, with the notable exception of B2 (related to academic management), in comparison to the other two in the same Dimension B, and to the rest in the remaining dimensions.

It was also found that the three 'check/act' criteria were more relevant globally, especially those relating to information management (**C1**). On the other hand, and consistent with the IRM dimensions, all planning criteria (**A1**: Design, **A2**: Personal, and **A3**: Resources) were depicted as causing influencers, along with one 'do' criterion—public information (**B3**) —and one 'check/act' criterion—information management (**C1**). In contrast, the criteria **B1**—'teaching–learning' and **B2**—academic management (related to the 'do' dimension) and, in smaller amounts, **C2**—'monitoring and evaluation of programmes' (related to the 'check/act' dimension) were clearly perceived as influenced criteria.

From matrix $[T_C]$, and following the steps in Equations (8)–(11), the unweighted supermatrix [W] (Equation (23)) was estimated.

| [W] | A1 | A2 | A3 | B1 | B2 | B3 | C1 | C2 | C3 |
|-----|-----|-----|-----|-----|-----|-----|-----|-----|-----|
| **A1** | 0.24 | 0.45 | 0.34 | 0.28 | 0.18 | 0.37 | 0.39 | 0.39 | 0.37 |
| **A2** | 0.30 | 0.19 | 0.41 | 0.29 | 0.36 | 0.33 | 0.26 | 0.29 | 0.25 |
| **A3** | 0.46 | 0.37 | 0.25 | 0.43 | 0.45 | 0.30 | 0.35 | 0.33 | 0.38 |
| **B1** | 0.51 | 0.52 | 0.48 | 0.34 | 0.59 | 0.48 | 0.48 | 0.45 | 0.41 |
| **B2** | 0.18 | 0.24 | 0.23 | 0.24 | 0.00 | 0.30 | 0.22 | 0.21 | 0.19 |
| **B3** | 0.31 | 0.24 | 0.28 | 0.42 | 0.41 | 0.22 | 0.30 | 0.35 | 0.39 |
| **C1** | 0.34 | 0.36 | 0.39 | 0.36 | 0.34 | 0.32 | 0.22 | 0.38 | 0.38 |
| **C2** | 0.35 | 0.37 | 0.33 | 0.34 | 0.36 | 0.39 | 0.43 | 0.26 | 0.40 |
| **C3** | 0.31 | 0.27 | 0.28 | 0.30 | 0.30 | 0.30 | 0.35 | 0.36 | 0.21 |

$$(23)$$

Subsequently, following the steps of Equations (12)–(15), the weighted supermatrix $[W_W]$ (Equation (24)) was determined by weighting the previous matrix $[W]$ with the transposed total relation matrix $[T_{D3}^{\alpha}]'$ of the three dimensions obtained in Equation (21).

| [Ww] | A1 | A2 | A3 | B1 | B2 | B3 | C1 | C2 | C3 |
|------|-----|-----|-----|-----|-----|-----|-----|-----|-----|
| **A1** | 0.04 | 0.08 | 0.06 | 0.07 | 0.05 | 0.09 | 0.13 | 0.12 | 0.12 |
| **A2** | 0.05 | 0.03 | 0.07 | 0.07 | 0.09 | 0.08 | 0.08 | 0.09 | 0.08 |
| **A3** | 0.08 | 0.07 | 0.04 | 0.11 | 0.11 | 0.08 | 0.11 | 0.11 | 0.12 |
| **B1** | 0.21 | 0.21 | 0.20 | 0.08 | 0.14 | 0.12 | 0.18 | 0.17 | 0.15 |
| **B2** | 0.07 | 0.10 | 0.09 | 0.06 | 0.00 | 0.07 | 0.08 | 0.08 | 0.07 |
| **B3** | 0.13 | 0.10 | 0.12 | 0.10 | 0.10 | 0.05 | 0.11 | 0.13 | 0.15 |
| **C1** | 0.14 | 0.15 | 0.16 | 0.18 | 0.17 | 0.16 | 0.07 | 0.12 | 0.12 |
| **C2** | 0.14 | 0.15 | 0.14 | 0.17 | 0.18 | 0.20 | 0.13 | 0.08 | 0.12 |
| **C3** | 0.13 | 0.11 | 0.12 | 0.15 | 0.15 | 0.15 | 0.11 | 0.11 | 0.06 |

$$(24)$$

Finally, by raising $[W_W]$ to high powers, the so-called 'global weights' of all criteria $[W_C]$ were obtained (Equation (25)); from these, the 'local weights' $[W_C^*]$ were calculated by dividing the global weights into those of their corresponding three dimensions $[W_{D3}]$.

| $[W_W]^n$ | $W_C$ (Global Weights) | | $W_{D3}$ | | $W_C^*$ (Local Weights) | $W_{D4}$ |
|-----------|------|------|------|---|------|------|
| **A1** | **0.09** | 0.09 | | | **0.34** | |
| **A2** | **0.08** | 0.08 | **0.26** | A | **0.30** | A |
| **A3** | **0.09** | 0.09 | | | **0.35** | |
| **B1** | **0.14** | 0.14 | | | **0.42** | |
| **B2** | **0.09** | 0.09 | **0.34** | B | **0.26** | B |
| **B3** | **0.11** | 0.11 | | | **0.33** | |
| **C1** | **0.14** | 0.14 | | | **0.34** | |
| **C2** | **0.14** | 0.14 | **0.40** | C | **0.35** | C |
| **C3** | **0.12** | 0.12 | | | **0.31** | |
| | | | | | **0.26** | D |

$$(25)$$

Both the local weights obtained for the criteria and those of the four dimensions (shaded in Equation (25)) were used to calculate the outputs of the rules in the FIS, as explained in the next section.

### 4.3. Fuzzy Inference System

The FIS proposed for evaluation was created in the likeness of the conceptual model and consisted of the following four subsystems. Three criteria subsystems (to evaluate the dimensions A—'plan', B—'do' and C—'check/act' from the valuations [0–10] of their corresponding criteria and a one-dimension subsystem (which collected the previous output evaluations, A, B and C, to be processed now as inputs, together with the valuation [0–10] given to the fourth dimension, D—'sustainability')

To design the inference subsystems of the model, the expert team agreed to use three labels for all input variables in the model (low—L, medium—M, high—H), five for the output variables of the criteria subsystems (very low—VL, low—L, medium—M, high—H, very high—VH) and seven for the output of the final dimension subsystem (extremely low—EL, very low—VL, low—L, medium—M, high—H, very high—VH, extremely high—EH), thus allowing a greater degree of discrimination in building the knowledge base. The process followed by the experts to determine a partition started with estimating their preferences regarding the potential location of each label core. Later, once the cores were agreed upon using a 2-tuple method [46], the complete labels were obtained by symmetry [45] in order to achieve strong partitions [50], which have better comprehensibility properties and satisfy important semantic constraints [51].

Figure 7 shows the agreed partitions of all model variables, the parameters of which are shown in Figure 8.

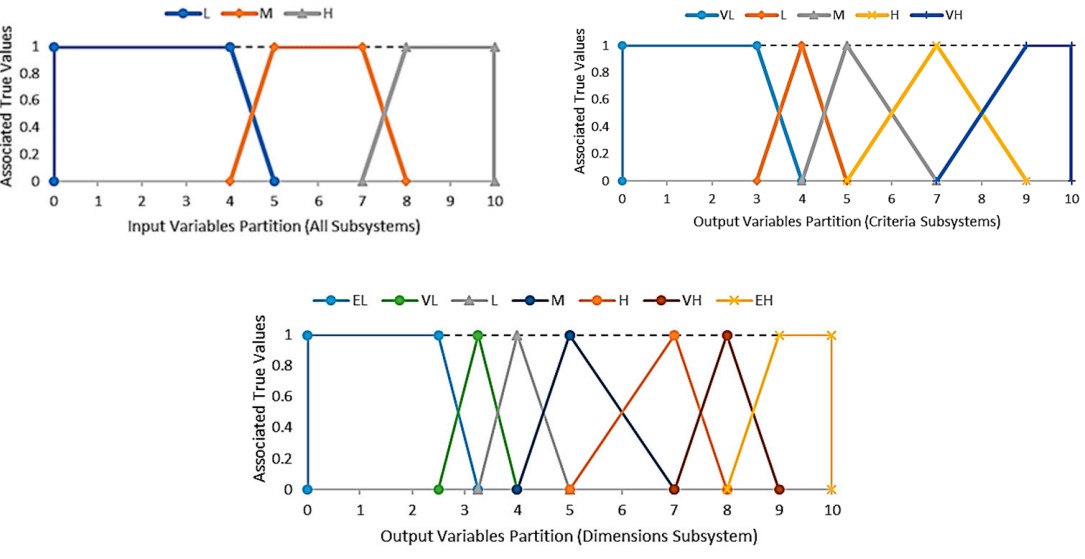

**Figure 7.** Agreed partitions of all model variables.

| Input (3) | | | | | Output (5) | | | | | Output (7) | | | |
|---|---|---|---|---|---|---|---|---|---|---|---|---|---|
| L | 0 | 0 | 4 | 5 | VL | 0 | 0 | 3 | 4 | EL | 0 | 0 | 2.5 | 3.25 |
| M | 4 | 5 | 7 | 8 | L | 3 | 4 | 4 | 5 | VL | 2.5 | 3.25 | 3.25 | 4 |
| H | 7 | 8 | 10 | 10 | M | 4 | 5 | 5 | 7 | L | 3.25 | 4 | 4 | 5 |
| | | | | | H | 5 | 7 | 7 | 9 | M | 4 | 5 | 5 | 7 |
| | | | | | VH | 7 | 9 | 10 | 10 | H | 5 | 7 | 7 | 8 |
| | | | | | | | | | | VH | 7 | 8 | 8 | 9 |
| | | | | | | | | | | EH | 8 | 9 | 10 | 10 |

**Figure 8.** Parameters of the labels in the model partitions.

Subsequently, by applying the methodology described in Section 3.2, and using the local weights of criteria and dimensions obtained in Section 4.2, the rule bases of the different subsystems were reached. Figure 9 shows an excerpt of the calculation of the first nine output rules of Inference Subsystem A (see last column). First, the output-weighted trapeze of each rule was calculated as the weighted sum of its input labels in A1, A2 and A3 (according to the weights $W_{A1} = 0.34$, $W_{A2} = 0.30$, $W_{A3} = 0.35$). Subsequently, the distances of each weighted trapeze to all potential output labels of a rule (VL, L, M, H, VH) were calculated. The final output label of each rule was the one corresponding to the minimum distance—shaded in this figure. Figure 10 shows the obtained rule bases of the model in the criteria subsystems.

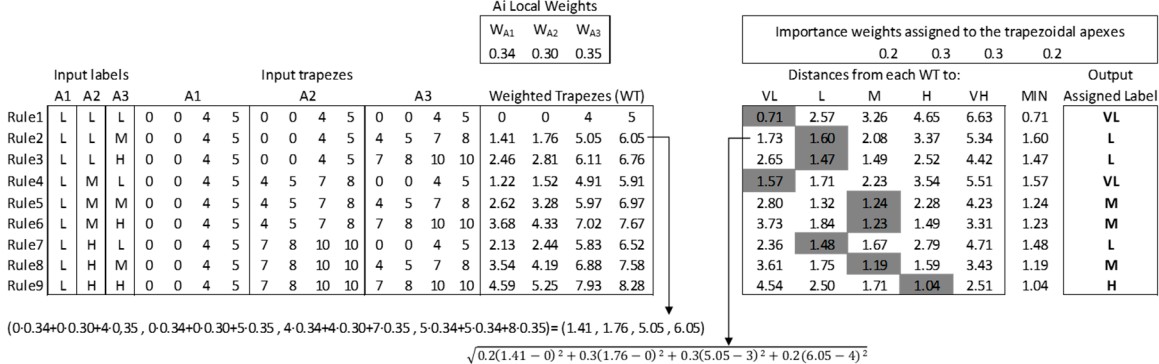

**Figure 9.** Calculation of the output labels for Subsystem A rules.

**Figure 10.** Rule bases in the criteria subsystems.

Once the knowledge base for all subsystems was designed, it was implemented in the MATLAB Toolbox Fuzzy Logic ©. Thus, it is possible to make inferences of the overall quality assessment of any institution in the EHEA according to the crisp values [0–10] granted to its nine criteria and to the D—'sustainability' dimension. Note that these crisp values should be quantified for each factor from their corresponding key performance indicators (KPIs), the ad hoc design of which is beyond the scope of this paper. In any case, the surface maps resulting from the different inference subsystems allow analysis of the evolution of both the assessments obtained in the different dimensions of the model and the overall quality assessment of any educational institution of the EHEA.

When interpreting the inference maps of any subsystem, it should be noted that each graph illustrates the output evolution of that subsystem for any value of two of its input factors (and for constant values of the factors not depicted in the chart). As an example, the maps obtained from the designed model are shown in Figures 11 and 12.

In all surface maps, a consistent evolution of the quality assessment of any institution can be observed. For increasing values of the variables illustrated in the base of the graph, progressively higher final evaluations can be obtained according to the surface shown, although with different nuances according to the analysed graph. On the one hand, the knowledge inserted in the rules of each subsystem (Figure 10) can prioritise some variables in the final evaluation—for example, in Figure 11 (right), quality is maximised for necessarily high values in A1, although not so high in A2. On the other hand, the different fixed values arranged for the variables not illustrated in the graph can substantially modify the shape of the surface maps (in general terms, the higher the fixed value, the greater the

height reached by the surface map, as can be seen in the successive maps of Figure 12). The results of all these maps are analysed in more detail below.

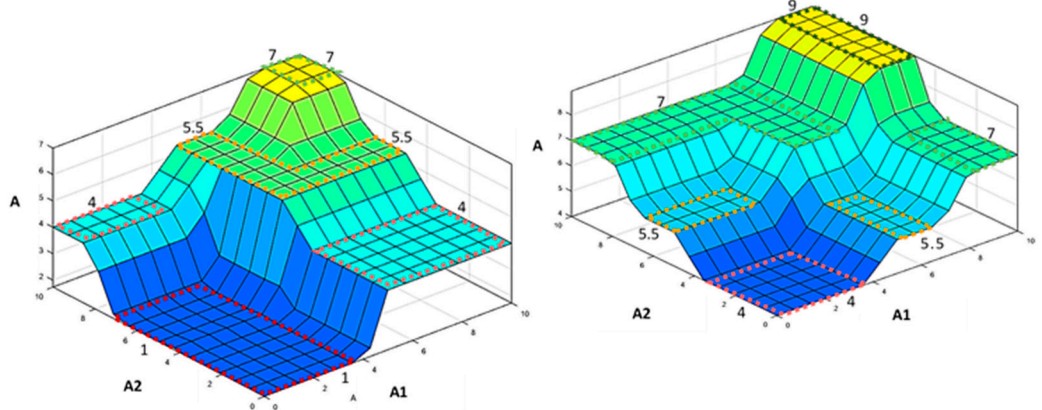

**Figure 11.** Dimension A inference map for low (left) and high (right) values of A3.

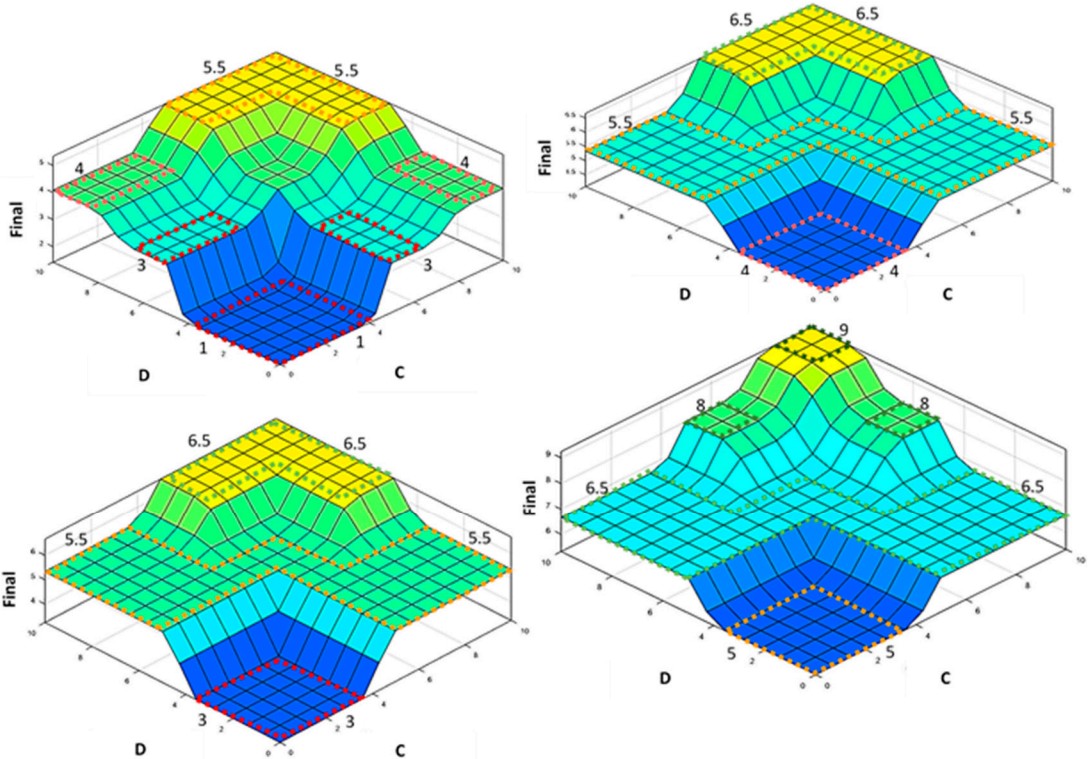

**Figure 12.** Assessment of the global quality (labelled as 'Final' on the Y-axis) according to Dimensions C—'check/act' and D—'sustainability' for different combinations of constant values of Dimensions A—'plan' and B—'do' (Top left: A low and B low; Top right: A low and B High; Bottom left: A High and B low; Bottom right: A High and B High).

Figure 11 shows the valuation maps of the A—'plan' dimension according to criteria A1 and A2. The map on the left corresponds to a 'low' value of A3 and the one on the right corresponds to a 'High' value.

In the left chart, the A—'plan' dimension started from a very low score of 1 for low A1 and low–medium A2 values. However, low A1 and high A2 values scored up to a 4 point valuation for A (as for medium–high A2 together with low A2 values). A total 5.5 points in A were reached for medium–high A1 and medium A2 values (as well as for medium A1 and medium–high A2 values).

Finally, by using high valuations in both A1 and A2, Dimension A gained the maximum score of 7 points.

In the right chart, Dimension A started from a low score of 4 for low values of A1 and A2. For low A1 and medium A2 values—and vice versa—Dimension A went up to 5.5 points. For mean values of A1 and A2, 7 points were reached (as well as for low–moderate A1 and high A2 values or high A1 and low A2 values). Finally, the maximum ratings of 9 was reached when A1 was high and A2 was medium–high.

The top-left map (Figure 12) was obtained for low values of A and B. When D and C were low, the global quality assessments were very low (1). When D was kept at low values but C was increased (and vice versa), the global quality value grew progressively in two steps until it reached a maximum of 4 points (which was also reached for intermediate values of D and C). Only with high values of D and medium–high values of C (and vice versa) was it possible to achieve medium global quality assessments of around 5.5 points.

The top-right map (Figure 12) was obtained for low values of A and high values of B. It started from low global quality assessments (4) for low values of the D and C dimensions. When D was kept at low values and C was increased (and vice versa) the global quality assessment grew progressively in only one step until it reached a maximum of 5.5 points. Only for high values of D and medium–high values of C (and vice versa) was it possible to achieve notable global quality values of around 6,5 points.

The bottom-left map (Figure 12) was obtained for high values of A and low values of B. The main difference with the previous map was the initial assessment of around 3 points.

The bottom-right map (Figure 12) was obtained for high values of A and B. It started from medium global quality assessments of 5 points when D and C were low. When D was kept at low values but C was increased (and vice versa) the global quality assessment grew progressively in only one step until it reached a maximum of 6,5 points (which was also reached for intermediate values of D and C). Only for high values of D and medium–high values of C (and vice versa) were excellent global quality assessments progressively achieved in two steps at around 8 and 9 points.

Nonetheless, in this latter subsystem, a rather symmetrical behaviour was noted in the four maps. This was due to the low dispersion of the model dimension weights ($W_A$-0.22; $W_B$-0.26; $W_C$-0.27; $W_D$-0.26) obtained according to the FDEMATEL methodology. These results highlight the similar importance of Dimension D dimension when compared to the other dimensions generally considered in such evaluations, and highlight the need to incorporate it into the overall quality assessment of EHEA.

The proposed method can be considered reliable since the results can be reproduced in an easy and traceable way by following the proposed methodology. To that end, a detailed explanation of the steps performed provided. With respect to the validity, the literature was reviewed in order to build a model that considers the existing theory and knowledge of the concept of quality to be measured and to detect other aspects, such us sustainability, that could be applied to enrich the model. However, given the novelty of both the model and the proposed methodology, comparison tests could not be carried out. In addition, to implement them, the development of KPIs would be required to initially quantify the crisp inputs of the model which, as already noted, exceeded the scope of this paper.

## 5. Conclusions

The assurance of quality and the achievement of excellence in higher education institutions has been and still is a global demand in current society. Accreditation as a mechanism to ensure better quality levels has emerged, however, as a continuous challenge, since its effectiveness has been questioned for different reasons that have been explained in the paper. This paper aimed to contribute to the quality assurance issue by first proposing a new conceptual framework based on the PDCA tool, wherein sustainability is considered a momentous dimension to be considered in the evaluation. Next, two MCDM techniques were used to establish priorities to be assigned to the model dimensions from the opinions provided by a group of experts, and to determine weights for the criteria encompassed

by those dimensions. Finally, a fuzzy inference system is built to make evaluations of educational institution quality based on the values of the model dimensions.

The proposed FIS allowed the building of the rule bases of the evaluation model by weighting the labels of their antecedents (from the weights obtained with the MCDM methodologies previously applied). This exempted the team of experts from reaching consensus on the output labels of the rules, avoiding possible inconsistencies with their influence assessments and saving resources in their determination.

The results showed a prominent role of by the sustainability dimension when explicitly included in the analysis, suggesting the need for a comprehensive revision of actual standards and guidelines so that sustainability principles are implemented and monitored in higher education quality evaluation procedures.

However, the results obtained in the design of the proposed model were limited by the knowledge provided by the expert team participating in the study. Therefore, a possible extension of this study would be to expand the number of interviewees on the panel of experts.

Other suggestions for future research include (a) the analysis and design of specific indicators for input crisp valuations (0–10) of both criteria inference subsystems and the sustainability dimension of the model; and (b) to identify, analyse and classify the key sustainability aspects in the evaluation of the institutions under study (based on the relevant literature in this area and the assessments that could be gathered from a panel of experts similar to the one proposed). In this way, these concepts could be incorporated into the determination of the indicators necessary to quantify the crisp valuations of the nine classic criteria set out in our model. In such a case, a model similar to the one proposed in this paper could be developed based exclusively on the nine criteria (already incorporating sustainability concepts) and the three initial dimensions: A—'plan', B—'do' and C—'check/act'.

**Author Contributions:** All authors have collaborated in the design of the model as well as in the collection of information, the analysis of the results and the establishment of conclusions. All authors have read and agreed to the published version of the manuscript.

**Funding:** This research received no external funding.

**Conflicts of Interest:** The author declare no conflict of interest.

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
