# Peer review of "Integrating Sustainability in the Quality Assessment of EHEA Institutions: A Hybrid FDEMATEL-ANP-FIS Model"

_sustainability, doi:10.3390/su12051707_

Round 1

Reviewer 1 Report

This manuscript proposes the design of a conceptual model of Quality Assessment in the European Higher Education Institutions. A Fuzzy Inference System is built to reach the evaluation of the Educational Institution Quality based on the values of the model dimensions. The results show the prominent role obtained by the sustainability dimension when explicitly included in the analysis. The inclusion of the sustainability as a necessary factor to be taken into account in the university context is suggested. I carefully read this paper, however, this manuscript need to be revised before it can be accepted. The following comments may help to improve the work of the authors.

General comments:

Several references cannot find, please revise all references again and follow the journal requirements. Technical writing of the manuscript needs to be improved, which several sentences are long and not clear. The step by step procedures of the hybrid methodology MCDM-FIS should be clarified. Author should clearly state how he integrated FDEMATEL with ANP-FIS model Results need to be discussed in a good manner, e.g., Figs. 10-11 should be clearly explained in details, otherwise, it is not clear why the variation of the parameters was observed in this way. To give a visual sense from the manuscript, a flowchart of the proposed model is suggested to be added. The author needs to show the benefit from this study and how we can benefit from this design of the conceptual model in the practical application?

Specific Comments:

Abstract:

The benefit from integrating Multicriteria Decision Making Approach- Fuzzy Inference System should be clarified. Results of this study needs to be clearer.

INTRODUCTION Many references are not defined e.g., Bucharest (2012) and Yerevan (2015), etc. What is the meaning of ESG? This sentence is not clear “Something similar happens in relation to the third pillar” Meaning is not clear; “however, also in this case, the criteria lack of the same emphasis and detail reflected in the first mission”. There are many recent publication on fuzzy decision making and risk assessment (e.g. http://dx.doi.org/1061/(ASCE)CO.1943-7862.0001757; https://doi.org/10.1016/j.scitotenv.2019.135310; https://doi.org/10.5194/hess-23-4293-2019; ), please refer to these publications. DIMENSIONS AND CRITERIA FOR THE QUALITY ASSURANCE IN HEI’S The observed tools for the utilization in the continuous quality improvement field such as PDCA tool -also referred to as the Deming cycle-, Six Sigma techniques, SERVQUAL model and EFQM model should be observed in details (https://doi.org/10.1016/j.scs.2019.101682; https://doi.org/10.1016/j.tust.2018.10.019). Author needs to state how he considered these observed tools in this study. METHODOLOGY This section needs to be observed again in a good manner. Author needs to state how he integrates FIS with The utilized parameters in this model are suggested to be added in Table. 3 needs to be clearly discussed. “From this matrix, a weighted supermatrix will be obtained through a normalization process (unit sum columns)”. Author needs to state how we can implement it. References of Eqs. 7-15 should be added.

 4. EMPIRICAL EVALUATION OF THE PROPOSED QUALITY ASSESSMENT MODEL. RESULTS AND DISCUSSION

9 which shows the excerpt from the calculation of the first 9 rules needs more discussion. 10-11: the discussion of this section is difficult to follow. The explanation of the Plan dimension inference map for a value of A3 low and high should be clarified in a simple way. Author needs to state the reason for giving novelty both in the model and the proposed methodology, comparison tests could not be carried out.

Author Response

RESPONSE TO REVIEWER 1 -– Sustainability

We thank you very much for the time you invested in giving us such as valuable and constructive feedback. This has greatly helped us to improve our article in this round of revision. In the following paragraphs, we explain how we have taken your comments into consideration.

GENERAL COMMENTS

All references in the paper have been reviewed, checking their correct citation and all of them appear in the bibliography. The paper has been submitted for professional review to improve the English language.

A methodology flow chart has been added (Figure 3) to improve its understanding. This diagram shows the merging between FDEMATEL and ANP methodologies. As stated in section 3.1.2., this hybridization avoids having to define a priori the structural interdependencies of the network, as well as, the usual consistency problems in the judgments given by experts, translated to the matrices of pairwise comparisons. In this way, the total relation matrix extracted from DEMATEL technique can be considered as the unweighted supermatrix in ANP, configuring a hybrid technique called DEMATEL-based ANP (DANP), which is the one used in this paper.

The discussion of the results has been enriched with a more general and intuitive explanatory paragraph that explains Figures 11 and 12. A more detailed explanation is also provided.

Regarding the benefits of the study, the proposed conceptual model itself allows to standardize the evaluation in HEIs, as evidenced in the conclusions section. The model is easily replicable for similar-to-the-proposal evaluation structures, which enables its practical application.

ABSTRACT

It has been shown that the use of this hybrid methodology allows to capture the existing influences among the criteria and dimensions of the model (FDEMATEL and FDANP) and the degree of subjectivity inherent to their evaluation processes (FIS)

INTRODUCTION

Some names followed by year in parenthesis were names of cities and not authors (e.g. Bucharest (2012), Yerevan (2015)). Hence, they were not cited in the references section. To avoid misunderstandings, we have used brackets instead of parentheses.

Although certain acronyms are quite well known in the scope of the study, the meaning of all of them has been arranged in their first appearance (e.g. "ESG")

The meaning of the three basic pillars in a university quality system: education, research and social impact has been clarified.

Relevant references have been added in the literature regarding both institutional quality assessment and, the use of the tools proposed in the development of the model. The suggested references move away from the proposed approach since they use different tools and in a different area such as risk assessment.

DIMENSIONS AND CRITERIA FOR THE QUALITY ASSURANCE IN HEIs

The aforementioned management tools (SERVQUAL, EFQM, PDCA, Six-sigma) are well-known tools in the field of continuous improvement and have been only mentioned for contextual purposes, their explanation exceeding the scope of our paper. However, the use of PDCA is justified by arguing its simplicity of understanding and adaptation to the analyzed problem.

METHODOLOGY

The flow chart of Figure 3 clarifies the steps of the research and the hybridization of methodologies. Section 3.2. details the calculation of the rule outputs in the FIS knowledge bases of the proposed evaluation model. These final rule outputs are obtained by the minimum distance among the potential labels assignable to the output variable and that obtained as a weighted sum (based on WC and WD) of the input labels of each rule.

The parameterization of the partitions of the model variables is obtained by consensus among experts and is presented for our specific example in Figures 7 and 8.

All references to paper equations have been added in the text.

EMPIRICAL EVALUATION

An explanatory paragraph related to the obtention process of the outputs of the rules was added before figure 9.

The explanation of figures 10 and 11 has been clarified.

Reviewer 2 Report

This is a very interesting paper which highlights extreme important issue on Integrating Sustainability in the Quality Assessment of EHEA Institutions.  The methods and models are relatively complex.

Here I must point out some main problems:

It is pointed to on page 4 that “sustainability” muss “as the fourth dimension (Dimension D) to be considered in the proposed conceptual model”. Nevertheless, it is not clear which indicators are used to measure “sustainability”. The authors use 7 pages to suggest a useful model to evaluate the quality of higher education. Nevertheless, “from Face-to-face interviews with EIGHT university members (referred to, from here on, as experts)” on page 6 indicates that there is very limited information collected, only through interviews from 8 experts. For a quantitative model, the results from a model with less than 30 samples is regarded as very unreliable. At the same time, it is not explained which kind of information is collected from 8 interviews. The missing descriptive results of the firsthand data leads to low reliability and validity.

Some more detailed suggestions:

Please adjust to the standard writing style and format, for instance: “(http://www.eees.es/)” in the text on page 1 should appear in footnote or endnote. The acronyms could show up for the first time after the whole word appear. For instance, “EHEA” and “FDEMATEL-ANP-FIS” should be followed by a footnote with to explain the whole word, so that the authors without knowledges in the same field could also understand the connotations of the acronyms. “the Fuzzy Decision Making Trial and Evaluation Laboratory method (FDEMATEL)” show up on page 5, although “FDEMATEL” already appears on page 1. Please use proofreading. For instance, “model is carry out” in the abstract, should be corrected as “model is carried out”. And “being education one of them” on page 4 should be revised as “education being one of them”. There are many more other mistakes in the text.

Author Response

RESPONSE TO REVIEWER 2 – Sustainability

We thank you very much for the time you invested in giving us such as valuable and constructive feedback. This has greatly helped us to improve our article in this round of revision. In the following paragraphs, we explain how we have taken your comments into consideration.

An extensive proofreading by a professional translation service was done. A methodology flow chart has been added (Figure 3) to improve its understanding. This diagram shows the merging between FDEMATEL and ANP methodologies. As stated in section 3.1.2., this hybridization avoids having to define a priori the structural interdependencies of the network, as well as, the usual problems of consistency in the judgments given by experts in the matrices of pairwise comparisons. In this way, the total relation matrix extracted from DEMATEL technique can be considered as the unweighted supermatrix in ANP, configuring a hybrid technique called DEMATEL-based ANP (DANP), which is the one used in this paper.

MAIN PROBLEMS

Sustainability Measurement. In the proposed model, “Sustainability” is only introduced as a dimension to be taken into account. The paper proposes the possibility of scoring this dimension on a 0-10 scale, although it is necessary to institutionally standardize the KPIs to be integrated into the mentioned qualification (this standardization does not fall within the scope of our paper and it is indicated as a possible research extension in the conclusions section).

Model unreliability. We have contacted more than 25 experts in institutional quality assessment to develop the study, although only 8 of them finally agreed to participate (we have noted this limitation in the final section of our study). It is really complex to find experts of this nature with sufficient experience, given the rotational character of this professional profile in higher education institutions. However, it is common to find relevant scientific studies in other areas that use this type of methodologies with expert panels of small dimensions (for instance, 3 experts in Abdel-Basset et al., 2018 or Büyüközkan & Güleryüz, 2016). Nevertheless, the proposed methodology is easily replicable and extrapolated to a greater number of experts.

The type of participation of experts in the study is now indicated in the sixth paragraph of the methodology: “In accordance with the above-mentioned techniques, the collaboration elicited from these university experts implied: a) to assess the interrelationship influences between each of the paired dimensions proposed in the model (from which the FDEMATEL method will be developed); b) to assess the interrelationship influences among each of the paired criteria proposed in the model (from which the FDANP method will be developed); c) to propose the partitions of the fuzzy variables of the model inference subsystems (once the partitions are agreed, the FIS can be further designed)”.

DETAILED SUGESTIONS

We have expanded the abstract explaining the meaning of acronyms related to the used methodologies and avoiding having to put footnotes (i.e. EHEA, FDEMATEL-ANP-FIS)

The paper has been sent to proofreading by a professional translation service to avoid grammar and spelling mistakes.

Round 2

Reviewer 1 Report

The quality of presentation is still poor. There are short paragraphs with only one or two sentences. Moreover, the results need to be compared with those from other method.

Author Response

RESPONSE TO REVIEWER 1 (ROUND2)

We thank you very much for the time you invested in giving us such as valuable and constructive feedback. This has greatly helped us to improve our article in this round of revision. In the following paragraphs, we explain how we have taken your comments into consideration.

The grammatical structure of the introduction section has been modified to avoid excessively short paragraphs. Additionally and in relation to the above, a certificate of the professional service provided by “Proof-Reading-Service.com” is attached, service which allowed for a substantial improvement in the writing of the paper.

In reference to the use of trapezoidal fuzzy numbers to be agreed upon by the expert team in the proposed methodology, we have included in Section 3.2. (Fuzzy Inference Systems) a relevant reference suggested by the reviewer, as it could complement our vision in achieving that consensus: “Hai-Min Lyu, Shui-Long Shen, Annan Zhou and Jun Yang. 2020. Risk assessment of mega-city infrastructures related to land subsidence using improved trapezoidal FAHP. Science of The Total Environment 28, 105007 ”

The nature of the presented paper is absolutely novel in the field of HEI evaluation processes (we have not found relevant bibliographic references that treat these processes in such a descriptive and traceable way as the one presented in the paper). In the first place, the novelty roots in the fact that there are no proposals for evaluation models incorporating sustainability dimensions in the evaluation of this type of institutions.

Secondly, we have merged several methodologies that are conveniently suited to this process by: 1) Proposing an evaluation model that combines the officially established criteria in this type of organizations (ESG standards) under the dimensions of the Deming cycle, together with a new dimension referred to Sustainability. 2) Formulating in deep detail the Fuzzy-DANP methodology, which allows both to avoid the potential inconsistency of judgments in the expert team and to add their paired “fuzzy” evaluations regarding the dimensions and criteria of the model. 3) Developing a fuzzy consensus inference system for the evaluation of the model in any HEI (incorporating proven techniques that allow to objectively agree both the fuzzy partitions of its variables and the outputs of their knowledge rule bases). 4) Illustrating, by means of surface maps, a rational behaviour in the evaluation inferred by all the subsystems of the model and the global system.

The innovative and novel nature of the proposed system prevents comparison of our results with those obtained from other methodologies, since these have not yet been developed in reference to the proposed model. However, our research group is firmly convinced that the proposed model will allow further studies to deepen the consideration of Sustainability criteria for the evaluation of HEIS with techniques similar to those described in the proposed work.

Round 3

Reviewer 1 Report

no comments

Author Response

Thank you very much for your recommendations.

All spelling and grammar controls requested have been addressed.

We hope that this latest version of our paper is now ready for publication.
